# Evaluation of Wild Peanut Species and Their Allotetraploids for Resistance against Thrips and Thrips-Transmitted Tomato Spotted Wilt Orthotospovirus (TSWV)

**DOI:** 10.3390/pathogens12091102

**Published:** 2023-08-28

**Authors:** Yi-Ju Chen, Sudeep Pandey, Michael Catto, Soraya Leal-Bertioli, Mark R. Abney, Sudeep Bag, Mark Hopkins, Albert Culbreath, Rajagopalbabu Srinivasan

**Affiliations:** 1Department of Entomology, University of Georgia, Griffin, GA 30223, USA; yijuchen@uga.edu (Y.-J.C.); sudeep.pandey@uga.edu (S.P.); 2Department of Entomology, University of Georgia, Athens, GA 30602, USA; mac65630@uga.edu; 3Department of Plant Pathology, Institute of Plant Breeding, Genetics and Genomics, University of Georgia, Athens, GA 30602, USA; sorayab@uga.edu; 4Institute of Plant Breeding, Genetics and Genomics, University of Georgia, Athens, GA 30602, USA; mark.hopkins@uga.edu; 5Department of Entomology, University of Georgia, Tifton, GA 31794, USA; mrabney@uga.edu; 6Department of Plant Pathology, University of Georgia, Tifton, GA 31793, USA; sudeepbag@uga.edu (S.B.); spotwilt@uga.edu (A.C.)

**Keywords:** host plant resistance, wild *Arachis*, spotted wilt of peanut, *Frankliniella fusca*, *Orthotospovirus*, vector fitness, diploid, induced tetraploid, thrips feeding index

## Abstract

Thrips-transmitted tomato spotted wilt orthotospovirus (TSWV) causes spotted wilt disease in peanut (*Arachis hypogaea* L.) and limits yield. Breeding programs have been developing TSWV-resistant cultivars, but availability of sources of resistance against TSWV in cultivated germplasm is extremely limited. Diploid wild *Arachis* species can serve as important sources of resistance, and despite ploidy barriers (cultivated peanut is tetraploid), their usage in breeding programs is now possible because of the knowledge and development of induced interspecific allotetraploid hybrids. This study screened 10 wild diploid *Arachis* and six induced allotetraploid genotypes via thrips-mediated TSWV transmission assays and thrips’ feeding assays in the greenhouse. Three parameters were evaluated: percent TSWV infection, virus accumulation, and temporal severity of thrips feeding injury. Results indicated that the diploid *A. stenosperma* accession V10309 and its derivative-induced allotetraploid ValSten1 had the lowest TSWV infection incidences among the evaluated genotypes. Allotetraploid BatDur1 had the lowest thrips-inflicted damage at each week post thrips release, while diploid *A. batizocoi* accession K9484 and *A. duranensis* accession V14167 had reduced feeding damage one week post thrips release, and diploids *A. valida* accession GK30011 and *A. batizocoi* had reduced feeding damage three weeks post thrips releasethan the others. Overall, plausible TSWV resistance in diploid species and their allotetraploid hybrids was characterized by reduced percent TSWV infection, virus accumulation, and feeding severity. Furthermore, a few diploids and tetraploid hybrids displayed antibiosis against thrips. These results document evidence for resistance against TSWV and thrips in wild diploid *Arachis* species and peanut-compatible-induced allotetraploids.

## 1. Introduction

Tomato spotted wilt orthotospovirus (TSWV) is the causal agent of the spotted wilt disease in peanut in the southeastern United States. Tomato spotted wilt orthotospovirus (*Orthotospovirus tomatomaculae*) is the type species in the genus *Orthotospovirus* and family *Tospoviridae*. The genome of TSWV consists of a tri-segmented negative or ambisense RNA molecules designated as S (small, 2.9 kb), M (medium, 4.8 kb), and L (large, 8.9 kb) enclosed within host-derived and double-layered membrane embedded with two glycoproteins [1]. TSWV is transmitted by several species of thrips in a persistent and propagative manner. The tobacco thrips, *Frankliniella fusca* (Hinds) (Thripidae: Thysanoptera), is the major vector of TSWV in peanut in southeastern United States [2,3,4,5]. The other well-known TSWV vector, *Frankliniella occidentalis* (Pergande), is also reported on peanut [6]. Besides transmitting TSWV, direct feeding injury of thrips may also contribute to economic losses in peanut production under certain circumstances [7]. 

TSWV was first documented in the United States in 1971 in Texas [8]. Since then, it has spread eastward and has constrained peanut production with yield loss peaking during 1997 at an estimated USD 40 million [9,10]. Georgia accounts for over 50% of the peanut acreage (800,000 acres). From 1996 to 2006, an annual average loss of USD 12.3 million due to TSWV was assessed in peanut production in Georgia [11]. Virus and/or vector-resistant cultivars were not available at the time of TSWV introduction into the United States, and management relied heavily on chemical and cultural tactics predominantly targeting the vector [12]. Simultaneously, a heavy emphasis was placed on breeding for resistance against TSWV and/or thrips. Since the 1990s, field-resistant cultivars against TSWV have been developed incrementally [12]. The first-generation cultivars, including ’Georgia Green’, were released in the 1990s [13]. The cultivars released in the last 20 years (second- and third-generation cultivars) are significantly more field resistant to TSWV than Georgia Green [12]. 

TSWV field resistance in peanut seems to be different from the typical hypersensitive response (HR) defined by local lesions induced by cell death at the site of virus inoculation [14,15,16]. The hypersensitive response (HR) on tomato and pepper are conferred by single genes, whereas evidence suggests that TSWV resistance in peanut resides in multiple chromosomes and that multiple genes could be involved in conferring field resistance. TSWV field-resistant cultivars are not immune to the virus and exhibit mild symptoms upon infection [17,18]. 

TSWV accumulation in some field-resistant cultivars seems to be lower than that of TSWV-susceptible cultivars [12,17,18]. The susceptibility of field-resistant cultivars also seems to vary with the thrips population and/or inoculum pressure, thereby indicating that the resistant cultivars cannot serve as ‘stand-alone’ management options. Chemical and cultural tactics are still used in conjunction with field-resistant cultivars to offset yield losses [6,12,19].

Given the status of field-resistant cultivars, peanut production would benefit by enhancing resistance. However, the genetic base within the cultivated peanut is narrow [20,21,22]. In the case of TSWV, the USDA accession PI203396, collected from Brazil in 1952, is the sole source of resistance for almost all TSWV-resistant cultivars released in the southeastern United States and has been estimated to have saved over USD 200 million a year [23]. However, this resistance has declined over time [10,12,21] and sources of resistance in the cultivated germplasm are very limited [24]. In contrast, wild *Arachis* species (diploid) have high genetic diversity and could be good sources of resistance against thrips and/or TSWV [24,25,26]. However, sexual incompatibility associated with ploidy-level differences and other fertility barriers present considerable difficulties in crossing with cultivated (tetraploid) peanut [24,26,27]. Induced tetraploid genotypes of *Arachis* spp. could help overcome these barriers and be amenable to breeding for resistance in cultivated peanut [25]. Thus far, host plant resistance derived from wild peanut genotypes against root-knot nematode (RKN, *Meloidogyne arenaria* (Neal)), late leaf spot (LLS, *Nothopassalora personata*), early leaf spot (*Passadora arachidicola*), and groundnut rosette-disease-associated viruses have been effectively introgressed into cultivated peanut [28,29,30,31,32]. 

The goal of this study was to evaluate diploid *Arachis* species and induced tetraploid hybrids as potential resistance sources against thrips and/or TSWV. An optimized thrips-mediated TSWV transmission assay was used as a high-throughput screening platform. Thrips feeding assays and fitness studies were conducted to assess the diploids and tetraploid hybrids as resistance sources against thrips.

## 2. Materials and Methods

### 2.1. Maintenance of Thrips Colonies

Nonviruliferous and viruliferous thrips (*F. fusca*) were maintained on detached peanut leaflets. Nonviruliferous thrips were collected in Georgia and established in 2009 at the University of Georgia and maintained on leaflets of noninfected plants (cv. Georgia Green) in Petri dishes with a wet cotton round. Colonies were maintained by successive releases of 10 adult female thrips, allowed to oviposit for 48 h on a peanut leaflet dusted with a trace of pine pollen as a supplement [33] and placed in growth chambers at 28–30 °C and a photoperiod of 14:10 (L:D). Fresh leaflets and water were added to the Petri plates three times a week until emergence of the F1 generation. The viruliferous thrips (TSWV) colony was maintained similarly on TSWV-infected leaflets collected from the field in a separate growth chamber as described in Shrestha et al. (2013) [34]. Partial sequences of several field-collected TSWV isolates are included in the Appendix A and deposited at NCBI GenBank (Accession numbers: OR482852-OR482908). During the off season, when possible, viruliferous thrips were maintained on TSWV-infected leaflets generated by mechanical inoculation, as described by Marasigan (2014), in the greenhouse [35].

### 2.2. Peanut Genotypes

Ten accessions of nine diploid *Arachis* species (*A. stenosperma* V10309, *A. duranensis* V14167, *A. cardenasii* GKP10017, *A. ipaënsis* K30076, *A. valida* GK30011 (PI 468154), *A. diogoi* V10602, *A. villosa* V12912, *A. batizocoi* K9484, *A. magna* K30092, and *A. magna* K30097; Table 1) and six induced allotetraploid hybrid genotypes of *Arachis* species (BatDur1, BatSten1, IpaVillo1, ValSten1, MagDur1, and MagSten1; Table 2) from the University of Georgia at Athens, GA, USA, were used for thrips-mediated inoculation in the greenhouse in 2019, 2020, and 2021. The common tetraploid cultivar ‘Georgia Green’ was used as a control in all inoculation assays. Seeds of diploids and tetraploid genotypes were treated with 2–3 mL of a 0.5% solution of Florel^®^ Growth Regulator (Monterey Lawn and Garden, Fresno, CA, USA) and incubated in a Petri plate at 28 °C for 18–24 h to break seed dormancy. Seeds were sown in individual four-inch pots with potting mix that included four parts sand, two parts field soil, two parts ProMix (Premier Horticulture Inc., Quakertown, PA, USA), and one part Perlite (Therm-O-Rock East Inc., New Eagle, PA, USA). Seeds of cultivar “Georgia Green” were pregerminated in moistened paper towel and incubated in a growth chamber kept at 28 °C for 2–3 days. Ten seedlings (ca. one to two week-old with one to two nodes and up to 16 leaflets) of each genotype were used for TSWV transmission. 

### 2.3. Thrips-Mediated TSWV Transmission

TSWV transmission was conducted via thrips-mediated inoculation [17]. For each genotype, 10 seedlings were placed in a thrips-proof cage (47.5 cm^3^) (Megaview Science, Taichung, Taiwan). Ten thrips (female adults up to three days old) from the viruliferous thrips colony were released on each seedling dusted with pollen as a supplement and enclosed within a plastic film cage (πr^2^ h = 3.14 × 16 × 39 cm^3^). TSWV detection was undertaken three weeks after thrips release.

#### 2.3.1. Incidence of TSWV Infection

The percentage of infected plants was estimated three weeks post inoculation. To determine TSWV infection status, plant tissue (ca 0.03 g) was collected from the first fully expanded leaf below the terminal and subjected to a double antibody enzyme-linked immune sorbent assay (DAS-ELISA). DAS-ELISA with TSWV-specific polyclonal antibodies (Adgia, Elkhart, IN, USA) was used to detect TSWV in peanut tissues in the laboratory using the protocol described in Lai et al., 2021 [38]. 

#### 2.3.2. Quantitation of Virus Loads in Infected Samples

The plants testing positive for TSWV by DAS-ELISA were selected for TSWV quantitation via a two-step reverse transcription–quantitative polymerase chain reaction (RT-qPCR) using TSWV-N gene-specific primers [39,40]. Symptomatic leaflet tissues (ca 0.03 g) from infected samples were used for RNA extraction. Total RNA was extracted by an RNeasy Plant Mini Kit (Qiagen, Valencia, CA, USA) and complementary DNA (cDNA) was synthesized by the Go-Script reverse transcription system (Promega Corporation, Madison, WI, USA) following the manufacturer’s instructions. Synthesized cDNA was used as a template for qPCR.

The reaction mix for qPCR included 12.5 μL 2× GoTaq qPCR Master Mix (Promega, Madison, WI, USA), forward and reverse primers (final concentration of 0.2 μM), 1 μL cDNA, and nuclease-free water for a final reaction volume of 25 μL. The reaction started at 95 °C for 2 min, followed by 40 cycles at 95 °C for 15 s, 58 °C for 60 s, and 72 °C for 20 s. The reaction was extended with melting curve analysis in a QuantStudio 3 system (applied biosystems by Thermo Fisher Scientific, Waltham, MA, USA) to rule out nonspecific binding. Each sample was tested in duplicate, and the absolute number of TSWV-N gene copies in the samples was quantitated using the standard curve protocol described by Shrestha et al., 2013 [34].

#### 2.3.3. Thrips Feeding Damage Index

Feeding injuries by nonviruliferous thrips were cataloged by the feeding damage index [18,41]. Feeding intensity (FI) was rated using a 0-to-3 scale. Foliage without feeding scars was rated as 0; foliage with less than 25% feeding scars was rated as 1; foliage with 25 to 50% feeding scars was rated as 2, and foliage with greater than 50% feeding scars was rated as 3 (Appendix A). Feeding damage (FD) indicates the proportion of leaflets with feeding scars (number of leaflets with feeding scars/total number of leaflets). Thrips feeding injuries were evaluated as feeding damage index (FDI, formula = FD × FI) at one, two, and three weeks post thrips release [41]. Each genotype had at least 10 plants. The experiment was conducted two times (*n* = 20 for each genotype). 

#### 2.3.4. Thrips Fitness

The effects of wild peanut genotypes on thrips fitness were evaluated by assessing thrips development time (adult to adult) and the number of adults produced. Ten diploid and six induced tetraploid genotypes were used for evaluating thrips fitness. Five Munger cages were set up for monitoring the fitness parameters [42]. Two leaflets (similar leaf size) of each genotype were placed in one Munger cage. Ten nonviruliferous female thrips (up to 3 days old) were transferred to leaflets using a fine brush for oviposition and removed after 72 h. Cages were kept in growth camber at 29 ± 1 °C with a 14 h:10 h (L:D) photoperiod. The dates of newly hatched larvae were recorded, and newly emerged adults were counted and removed daily. The experiment was repeated once (*n* = 10 Munger cages for each genotype). 

#### 2.3.5. Correlation between Final TSWV Infection and Thrips Feeding Injury

To assess the correlation between feeding injury at one, two, and three weeks post inoculation and TSWV infection (infection percentage at three weeks post inoculation), data from all the experiments on wild diploid and tetraploid genotypes were pooled. The comparisons were made by Pearson’s product-moment correlation ‘cor.test’ at a 95% confidence interval in software R version 4.1.0 [43]. 

### 2.4. Statistical Analyses

Statistical analysis was performed to compare TSWV infection between wild diploid species and their tetraploid hybrids with Georgia Green. The data from two experimental repeats were pooled (*n* = 20 for each genotype) and subjected to generalized linear mixed-model analysis using the “glmer” function in software R, assuming a binomial distribution using the logit link function [43]. The TSWV-N gene copies between diploid species and their tetraploid hybrids with Georgia Green were analyzed using the “glmer” function (the generalized linear mixed-model analysis, GLMM) in software R version 4.1.0 with Gaussian distribution [43]. The observations over time for the thrips damage index were analyzed separately for each time point. Data from all the replications were pooled. Thrips feeding injury data were subjected to GLMM analysis using the “glmer” function in software R assuming a normal distribution [43]. Thrips fitness data (number of adults produced) were subjected to GLMM analysis using the “glmer” function in software R version 4.1.0 assuming a Poisson distribution. Genotypes (treatments) were considered as fixed effects, and replications and experimental repeats were considered as random effects. Least square means (LS-mean) was used for multiple comparisons at a significance level of α= 0.05 with Tukey adjustment to determine the difference between genotypes using the “lsmeans” function in R. The median developmental time required for the adults to develop was evaluated on each genotype, and the statistical significance of the difference between genotypes was estimated by a Kruskal–Wallis test using the “kruskal.test” function in R version 4.1.0 [43]. 

## 3. Results

### 3.1. TSWV Symptoms on Diploids and Allotetraploids

All wild diploid and tetraploid genotypes showed visible systemic symptoms upon TSWV infection (Figure 1). The common symptoms included ringspots, chlorotic spots, yellowing, necrosis, wilting, stunting, and death of terminal buds. Diverse symptoms were observed on each genotype and varied with disease progression. Most wild peanut genotypes showed concentric ringspots on leaflets and continued to grow, as in the case of ValSten1. However, the plants with dry terminal buds resulted in the death of plants such as *A. duranensis* V14167 and *A. ipaënsis* K30076. 

### 3.2. Incidence of TSWV Infection

TSWV infection percentage varied within wild genotypes of the *Arachis* species (F_16,697_ = 4.6332; *p* < 0.001; Figure 2). Diploids, *A. stenosperma* V10309 (Sten, 22.0 ± 7.3%), were infected at a reduced percentage than the cultivated genotype *A. hypogea* cv. Georgia Green, which was used as the control genotype in each experiment. The infection percentages in *A. villosa* V12912 (Villo, 26.1 ± 17%), *A. diogoi* V10602 (Dio, 29.4 ± 12.0%), *A. ipaënsis* K30076 (Ipa, 36.0 ± 3.4%), *A. valida* GK30011 (Valida, 37.6 ± 14.7%), *A. cardenasii* GKP10017 (Car, 26.7 ± 8.8%), and *A. batizocoi* K9484 (Bat, 44.3 ± 9.1%) were lower but not significantly different from *A. hypogea*. TSWV infection percentages in *A. duranensis* V14167, *A. magna* K30097, and *A. magna* K30092 were higher than that of *A. stenosperma* V10309. In allotetraploids, the TSWV infection percentage in ValSten1 (14.1 ± 4.1%) was significantly lower than the control genotype (Georgia Green, 65.5 ± 3.1%). 

### 3.3. Quantitation of Virus Loads in Infected Samples

Virus loads in terms of TSWV-N gene copies were compared between diploids and tetraploids along with Georgia Green by RT-qPCR. The results showed differences between genotypes examined among diploids (F_10,66_ = 1.91, *p* = 0.0002; Figure 3A) and tetraploids (F_6,54_ = 0.64, *p* = 0.008; Figure 3B). However, TSWV loads in diploids and/or tetraploids were not significantly different to those of the control genotype (Figure 3). In diploids, TSWV loads in *A. villosa* V12912, *A. batizocoi* K9484, and *A. magna* K30092 were significantly higher than in *A. cardenasii* GKP10017, *A. valida* GK30011, and *A. duranensis* V14167 (Figure 3A). In induced tetraploid genotypes, TSWV loads in MagSten, BatSten1, and BatDur1 were significantly higher than in MagDur1 (Figure 3B).

### 3.4. Thrips Feeding Damage Index

In diploid genotypes, feeding injury differences were observed at one week (F_10,397_ = 16.94; *p* < 0.0001; Figure 4A), two weeks (F_10,423_ = 16.01; *p* < 0.0001; Figure 4B), and three weeks (F_10,419_ = 23.57; *p* < 0.0001; Figure 4C) post thrips release. More feeding injuries were recorded in *A. magna* K30097 alone than the cultivated genotype (Georgia Green) at one week after thrips release, and *A. batizocoi* K9484 and *A. duranensis* V14167 had significantly fewer feeding scars than the control genotype (Figure 4A). Also, every diploid, except *A. diogoi* V10602, at three weeks post thrips release, showed fewer thrips injuries than *A. magna* K30097 (Figure 4A–C). 

In tetraploid genotypes, feeding injury differences were observed at one week (F_6,270_ = 14.39, *p* < 0.0001; Figure 4D), two weeks (F_6,293_ = 7.51, *p* < 0.0001; Figure 4E), and three weeks (F_6,272_ = 15.36, *p* < 0.0001, Figure 4F) after thrips release. One week after thrips release, thrips feeding injuries were lower in BatDur1 and higher in both MagSten and IpaVillo1 than in the control genotype, Georgia Green (Figure 4D). Two weeks after thrips release, only BatDur1 had fewer thrips feeding scars in comparison with the control genotype (Figure 4E). Three weeks after thrips release, ValSten1, BatDur1, and BatSten1 had less thrips injury than the control genotype (Figure 4F).

The thrips thrips injury increased with time except genotype BatSten1 (Figure 5). BatSten1 showed substantial defoliation after thrips injury. Therefore, there were fewer and less severe feeding scars observed on BatSten1 at three weeks than at two weeks after feeding. Also, the defoliation of lower leaves was observed on *A. stenosperma* V10309, *A. valida* GK30011, and MagSten1.

### 3.5. Thrips Fitness

The survival of thrips was estimated by the number of adults emerging from each cage (10 females released per cage). The number of adults varied between genotypes in diploids (F_10,105_ = 134.35, *p* < 0.0001; Figure 6A) and tetraploids (F_6,53_ = 79.005, *p* < 0.0001; Figure 6B). The survival of thrips on diploids was five times lower than in the control, Georgia Green. Similar to the diploids, the number of adults that emerged on tetraploids was lower than in the control. Within the tetraploid genotypes, thrips survival was the highest in MagSten1 and lowest in BatDur1.

The median developmental time of *F. fusca* ranged from 13 to 16 days on 17 *Arachis* accessions at 29 °C. Developmental time varied within diploids (x^2^ = 43.2, df = 10, *p* < 0.0001; Figure 7A) and within tetraploids (x^2^ = 41.6, df = 6, *p* < 0.0001; Figure 7B). The median developmental time was shorter on the cultivated genotype ‘Georgia Green’ (12.8 d) than on diploids and tetraploids. The median developmental time was the shortest and longest on *A. villosa* V12812 (14.3 d ± 0.26) and *A. ipaënsis* K30076 (15.9 d ± 0.27), respectively. In tetraploids, the median developmental time was the shortest and longest on BatSten1 (13.7 d ± 0.40) and MagDur1 (16.0 d ± 0.20), respectively.

### 3.6. Correlation between Final Percent TSWV Infection and Thrips Feeding Injury 

Seven days post thrips-mediated inoculation, no correlation between thrips feeding and percentage of TSWV infection was observed with diploids (y = 0.1428x + 0.3061, R^2^ = 0.047, Correlation = 0.216, *p* = 0.2345; Figure 8) and tetraploids (y = −0.03692x + 0.60439, R^2^ = 0.004, Correlation = −0.06, *p* = 0.7245; Figure 8). While thrips feeding injury recorded after two weeks showed a positive correlation in the case of diploid genotypes (y = 0.1736x + 0.1813, R^2^ = 0.167, Correlation = 0.408, *p* = 0.01832; Figure 8), a positive correlation was not observed with tetraploids (y = 0.09238x + 0.40721, R^2^ = 0.097, Correlation = 0.312, *p* = 0.094; Figure 8). After three weeks of feeding, a positive correlation between thrips feeding injury and TSWV transmission was observed with both diploids (y = 0.221x + 0.02921, R^2^ = 0.366, Correlation = 0.605, *p* = 0.00011; Figure 8) and tetraploids (y = 0.1840x + 0.1240, R^2^ = 0.268, Correlation = 0.518, *p* = 0.002; Figure 8).

## 4. Discussion

Thrips-mediated inoculation of TSWV indicated that some diploids such as *A. stenosperma* V10309 (AA genome) were infected at a lower level than other diploids and the cultivar, Georgia Green. *Arachis stenosperma* V10309 has been shown in earlier studies to possess resistance against fungal pathogens and root-knot nematode (RKN) [44,45,46]. Among the induced allotetraploids evaluated, ValSten1 had a reduced TSWV infection percentage (~30%) in comparison with others. An earlier field study also documented ValSten1 to be tolerant to TSWV compared with the susceptible cultivars ‘Gregory’ and ‘Florunner’ [29]. Also, ValSten1 was documented to exhibit resistance to early and late leaf spot and rust pathogens [37]. These traits, in addition to TSWV resistance, make this genotype promising for peanut breeding, and it also is vigorous and highly fertile [37]. Some of the results observed in this study were in contrast with earlier findings; for instance, *A. diogoi* V10602 did not display higher levels of TSWV resistance and exhibited TSWV symptoms upon inoculation. Previously, *A. diogoi* V10602 showed no symptoms following TSWV inoculation [27]. This could be because Lyerly et al. (2002) used mechanical inoculation [27], whereas this study used thrips-mediated inoculation. Thrips-mediated inoculation has shown to be more effective and consistent than mechanical inoculation [17]. Lai (2015) also used thrips-mediated inoculation and demonstrated that *A. diogoi* V10602 accession can get infected with TSWV [47]. 

Despite the reduced TSWV infection percentages in some diploids and tetraploids when compared with Georgia Green, TSWV accumulation levels were not congruent to the infection percentages. For instance, the genotypes, *A. cardenasii* GKP10017, *A. valida* GK30011, and *A. duranensis* V14167 were infected at higher percentages but accumulated reduced virus loads when compared with other diploids and tetraploids. There was no evidence of hypersensitive response against TSWV in any of the genotypes evaluated in this study. Instead, all diploid and tetraploid genotypes displayed systemic TSWV symptoms. Most wild species developed the same characteristic TSWV symptoms, such as ringspots, yellowing, and stunting, as in the case of peanut cultivars, but ringspots were not observed in the case of a few genotypes such as *A. villosa* V12812. Overall, the severity of TSWV symptoms on the evaluated genotypes was somewhat milder than in the susceptible cultivar, Georgia Green. However, the dry terminal bud was a severe TSWV symptom in a few wild peanut genotypes. Given that seedlings of wild peanut genotypes often have one or very few terminal bud(s) compared with cultivated peanut, the drying of terminal buds could be lethal. TSWV-induced phenotypes in this study could potentially be influenced by the different isolates (from field-collected foliage). Partial sequencing of TSWV isolates from the last 10 years, including the isolates used in this study, indicated a tremendous amount of variation in sequences in a given spatial scale [18,48]. However, purifying selection seems to be the major factor influencing the population genetics and evolution. Also, isolate effects on the host phenotype were not influenced by host resistance to TSWV [18,48]. Despite using multiple isolates in this study, the phenotype induced by TSWV does not seem to be affected by host resistance in diploids. Also, it should be noted that Georgia Green (a TSWV-susceptible cultivar) was used as a control in every transmission assay with diploids and allotetraploids.

TSWV-induced phenotypes observed on *Arachis* diploids and induced allotetraploids suggest that their mechanism of TSWV resistance is similar to what has been observed in the case of cultivated tetraploids obtained via conventional plant breeding [17,18]. The field resistance to TSWV in peanuts has been found to be associated with more than one quantitative trait loci [49,50,51] and suggests that the resistance against TSWV in peanut is different from the single-gene-governed resistance observed in the case of solanaceous crops such as tomato and pepper [15,52]. The results from this study with diploids and induced tetraploids together also supported the possibility of the involvement of multiple genes associated with TSWV resistance in peanut [53,54,55].

Previous studies have indicated that in addition to reduced TSWV infection in some of the field-resistant cultivars, there also was enhanced resistance and/or tolerance against tobacco thrips, *F. fusca* [12]. Thrips feeding damage was evaluated in this study. Diploids such as *A. batizocoi* K9484 and *A. duranensis* V14167, as well as tetraploids such as BatDur1, had reduced thrips feeding-induced damage when compared with the susceptible genotype, Georgia Green, two and three weeks post thrips infestation. A previous study also provided evidence for reduced thrips susceptibility in *A. batizocoi* K9484 (=PI298639) and PI468329 [56,57]. Overall, less feeding injury was recorded in the diploids than in the induced tetraploids. Similarly, Michelotto et al. (2017) observed that certain allotetraploids showed higher susceptibility to another thrips species (*Enneothrips flavens* = *E. enigmaticus*) than its parental diploids [58].

Reduced thrips feeding injuries in some diploids and a few tetraploids could be due to antixenosis (nonpreference). Morphological leaf traits can be the first physical barrier against thrips and influence nonpreference. For instance, the types and density of trichomes differentially affected thrips resistance in strawberries and wild tomato [59]. Wild diploid peanut genotypes have been documented to possess higher trichome density than allotetraploids [36]. Therefore, it is possible that the thrips resistance to diploids may be correlated with trichome density in this study as well. Additionally, antixenosis can be associated with chemicals/metabolites and needs to be explored with the evaluated diploids and induced allotetraploids [60]. In addition to antixenosis, antibiosis (negative effects on thrips fitness) is another factor that could be responsible for resistance against thrips. The fitness experiments conducted under controlled conditions showed that the diploid genotypes of wild *Arachis* species were unsuitable hosts for thrips, as many individuals were not able to complete their life cycle. The thrips survival percentage was consistently lower in diploid genotypes than in the others evaluated in this study. In tetraploid genotypes, the pattern of adult emergence was consistent with that of diploids, but in some instances, there was reduced heritability of resistance/tolerance traits in tetraploids when compared with parental diploids. For example, MagDur1 was more susceptible to thrips than both parental genotypes. The thrips resistance trait of ValSten1 was likely inherited from the AA genome progenitor (Sten1). Overall, thrips on diploids and tetraploids took a longer time to complete one generation than on Georgia Green.

In this study, several diploids and induced tetraploids were infected at a reduced percentage versus others, and a few of them also exhibited resistance/tolerance against thrips. However, the correlation between TSWV and thrips resistance was not substantial. This indicated that TSWV and thrips resistance are not governed by the same set of genes. The absence of correlation between TSWV and thrips resistance in several diploids and tetraploids poses a significant challenge for plant breeders. In this study, only nine out of the many extant *Arachis* species were evaluated. Other unevaluated diploid species could also serve as TSWV-resistance sources and remain to be explored. 

## 5. Conclusions

TSWV is one of the most serious pathogens that infect the peanut crop. The development of TSWV-resistant cultivars has been a vast improvement. Despite the usage of these resistant cultivars, losses continue to occur in proportion to the thrips and TSWV inoculum pressure. Almost all currently available peanut cultivars in the southeastern United States have been generated from a single accession with TSWV resistance. With such a narrow genetic base, there is room for evolution of resistance-breaking variants, as noticed in other crops [14,52,61]. Recent research provided no evidence for high levels of host resistance-induced virus selection pressure against TSWV in peanut in Georgia, USA [48]. However, intense cultivation of peanut in over a million acres in the southeastern United States annually might facilitate the development of resistance-breaking strains in the future. Broadening the genetic base of TSWV resistance with induced allotetraploids (derived from diploids) might mitigate and/or thwart the development of highly virulent and/or resistance-breaking TSWV strains. 

## Figures and Tables

**Figure 1 pathogens-12-01102-f001:**
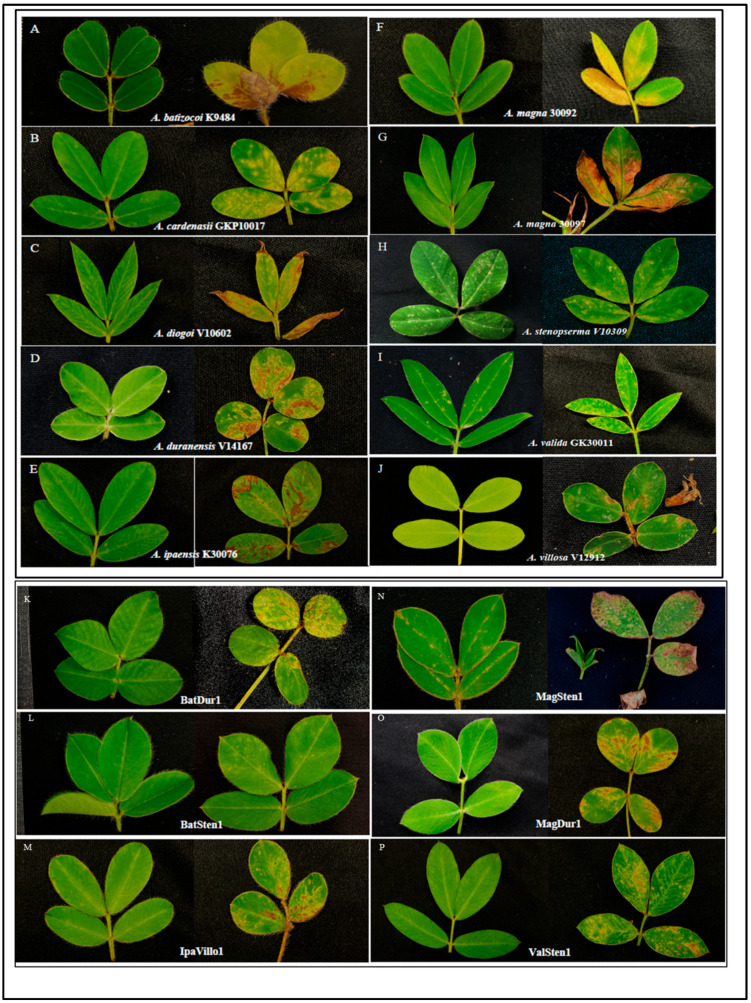
Symptoms of tomato spotted wilt orthotospovirus (TSWV) infection on leaflets of diploid (**A**–**J**) and allotetraploid (**K**–**P**) *Arachis* genotypes. Noninfected leaflet (**left**) and TSWV-induced symptoms (**right**) on diploid and allotetraploid *Arachis* genotypes. The pictures were taken approximately three weeks post inoculation.

**Figure 2 pathogens-12-01102-f002:**
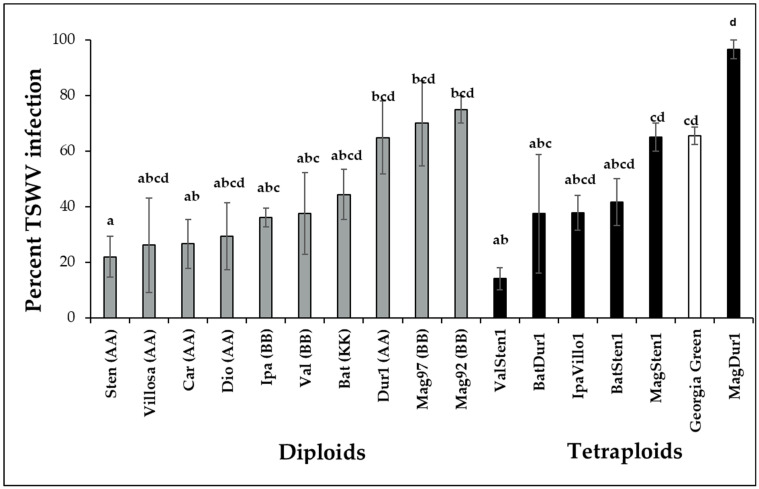
Mean percentage infection of tomato spotted wilt orthotospovirus virus (TSWV) in wild species of *Arachis* genotypes with AA, BB, or KK genomes and their induced allotetraploids genotypes three weeks post inoculation. Ten potentially viruliferous tobacco thrips were used to inoculate a single plant. Ten plants were tested for each genotype, and the experiment was repeated twice (*N* = 20). Cultivated tetraploid (*A. hypoaea* cv. Georgia Green) was used as the control for each experiment. The infection status of inoculated plants was determined by double-antibody sandwich enzyme-linked immunosorbent assay (DAS-ELISA) targeting the nucleocapsid protein (N) of TSWV. Different letters on standard error of means (SE) indicate significant differences between means separated by LSD with the Tukey method at α = 0.05.

**Figure 3 pathogens-12-01102-f003:**
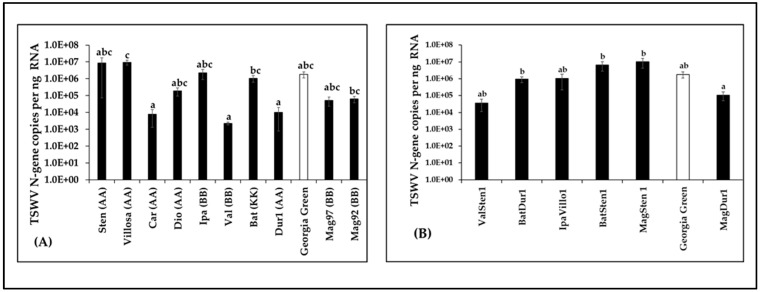
Tomato spotted wilt orthotospovirus (TSWV) accumulation in *Arachis* genotypes three weeks post inoculation. Virus loads of TSWV from infected leaflet samples in wild diploids (**A**), induced allotetraploids (**B**), and cultivated peanut (*A. hypoaea* cv. Georgia Green, indicated by white bar in the figure) were estimated by reverse transcription–quantitative polymerase chain reaction followed by absolute quantitation using plasmids containing TSWV N-gene inserts as standards. Different letters on standard error of means (SE) indicate significant differences between means separated by LSD with the Tukey method at α = 0.05.

**Figure 4 pathogens-12-01102-f004:**
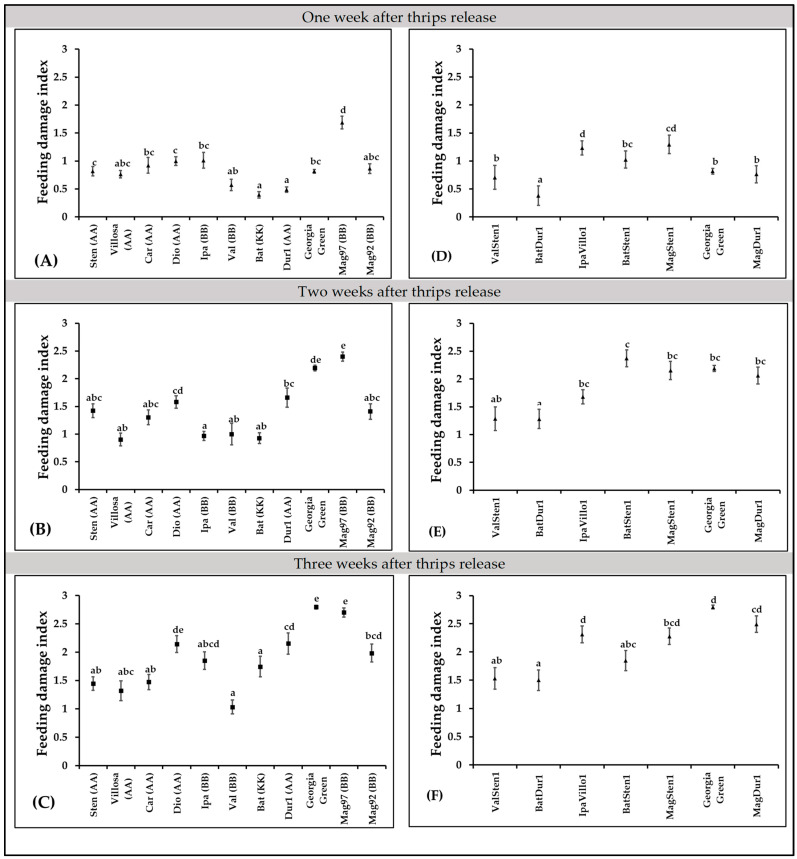
Thrips feeding injury on *Arachis* genotypes. Feeding damage indices (FDI) were evaluated at one week, two weeks, and three weeks post thrips release on wild diploids (**A**–**C**) and induced allotetraploids (**D**–**F**). Cultivated tetraploid (*A. hypoaea* cv. Georgia Green) was used as the control at each experiment. Mean feeding damage indices are presented in one-week intervals. Different letters on standard error of means (SE) indicate significant differences between means separated by LSD with the Tukey method at α = 0.05.

**Figure 5 pathogens-12-01102-f005:**
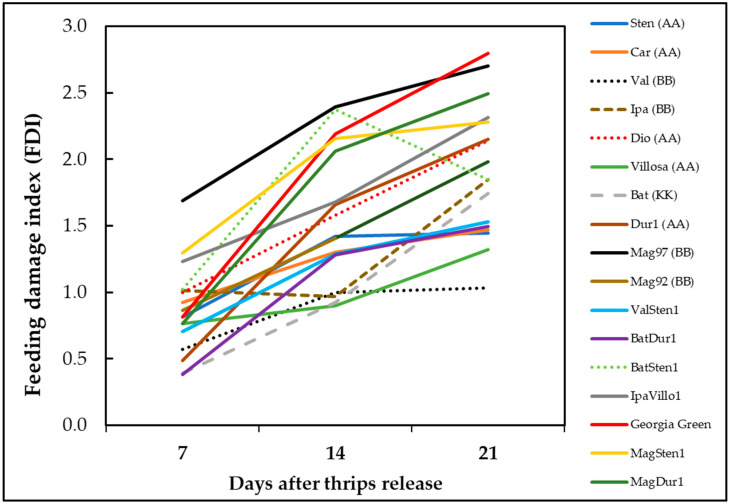
Accumulation of thrips feeding injury on *Arachis* genotypes over time. Feeding damage indices (FDI) were evaluated at one week, two weeks, and three weeks post thrips release on wild peanut genotypes. Cultivated tetraploid (*A. hypoaea* cv. Georgia Green) was used as the control at each experiment.

**Figure 6 pathogens-12-01102-f006:**
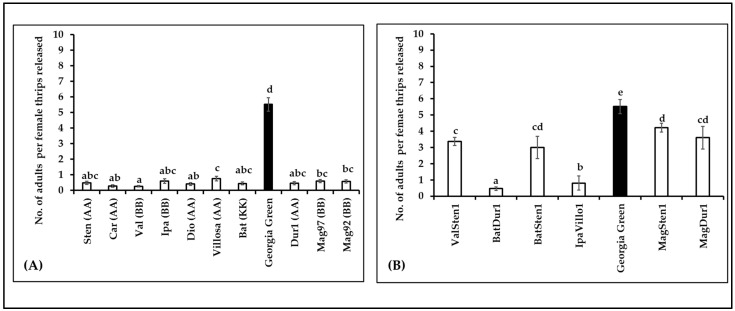
Mean number of adults emerging per *Frankliniella fusca* female thrips released on the leaflets of noninfected *Arachis* genotypes ((**A**): diploids, (**B**): tetraploids). Ten nonviruliferous female thrips were released on noninfected leaflets on each Munger cage. Approximately 10 Munger cages were set up for each genotype. The number of adults that emerged from each cage was recorded at 24 h intervals. Different letters on standard error of means (SE) indicate significant differences between means separated by LSD with the Tukey method at α = 0.05. The lowest number of adult thrips emerging per female were observed in Val and Batdur1 among diploids and tetraploids, respectively. Georgia green (indicated as black bar in the figure) was used as a control.

**Figure 7 pathogens-12-01102-f007:**
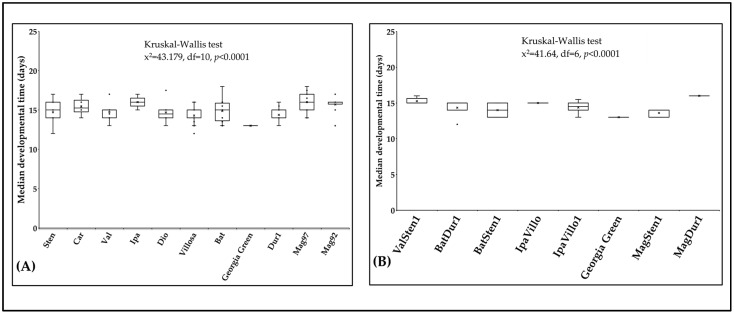
Median developmental time of *Frankliniella fusca* on plants. The time required for thrips to complete one generation on leaflets of noninfected *Arachis* genotypes ((**A**): diploids, (**B**): tetraploids) was recorded daily. The ‘horizontal line’ and ‘X’ in the box represent median and mean of median developmental time, respectively.

**Figure 8 pathogens-12-01102-f008:**
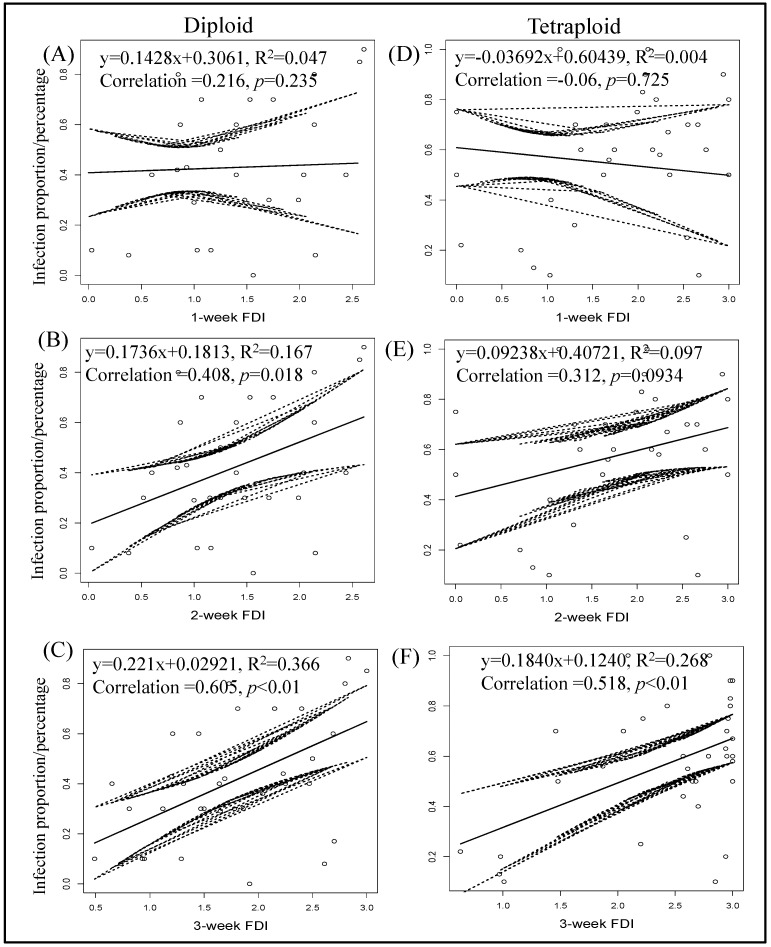
Correlation between final percent infection of tomato spotted wilt virus and thrips feeding injury by time. Feeding scars produced on each plant were recorded three times at one-week intervals during the thrip-mediated inoculation. Pooled data from all experiments on wild diploids (**A**–**C**) and tetraploid (**D**–**F**) genotypes are presented. Equations of linear regression and R square are shown.

**Table 1 pathogens-12-01102-t001:** Wild diploid *Arachis* genotypes used for thrips-mediated inoculation.

Wild Diploid	Plant ID	Accession No. (USDA No.)	Genome Type	Collection Site
*A. batizocoi*	Bat	K9484 (PI 298639)	KK	Parape, Bolivia
*A. cardenasii*	Car	GKP10017 (PI262141)	AA	Robore, Bolivia
*A. diogoi*	Dio	V10602 (PI 276235)	AA	Paraguay
*A. duranensis*	Dur1	V14167	AA	Salta, Argentina
*A. ipaënsis*	Ipa	K30076 (PI 468322)	BB	Gran Chao, Bolivia
*A. magna*	Mag92	K30092 (PI 468337)	BB	Bolivia
*A. magna*	Mag97	K30097 (PI468340)	BB	Santa Cruz, Bolivia
*A. stenosperma*	Sten	V10309 (PI 666100)	AA	Mato Grosso, Brazil
*A. valida*	Val	GK30011 (PI 468154)	BB	Mato Grosso, Brazil
*A. villosa*	Villo	V12812	AA	Bella Union, Uruguay

**Table 2 pathogens-12-01102-t002:** Wild induced allotetraploid *Arachis* genotypes used for thrips-mediated inoculation.

Induced Allotetraploids	Plant ID	Genome Type	Collection/Registration
*A. batizocoi* K9484 × *A. duranensis* V14167	BatDur1	KKAA	[20]
*A. batizocoi* K9484 × *A. stenosperma* V10309	BatSten1 (PI 695418)	KKAA	[20]
*A. ipaënsis* K30076 × *A. villosa* V12812	IpaVillo1	BBAA	[36]
*A. magna* K30097 × *A. duranensis* V14167	MagDur1	BBAA	Unpublished
*A. magna* K30097 × *A. stenosperma* V10309	MagSten1 (PI 695417)	BBAA	[36]
*A. valida* GK30011 × *A. stenosperma* V10309	ValSten1 (PI 695393)	BBAA	[29,37]

## Data Availability

Partial TSWV gene sequences from this study have been deposited at a public repository: NCBI GenBank accession numbers-OR482852-OR482908.

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
