# Peer review of "Evaluation of Wild Peanut Species and Their Allotetraploids for Resistance against Thrips and Thrips-Transmitted Tomato Spotted Wilt Orthotospovirus (TSWV)"

_pathogens, 2023, doi:10.3390/pathogens12091102_

Round 1
Reviewer 1 Report
Title: Evaluation of Wild Peanut Genotypes for Resistance Against Thrips and Thrips-Transmitted Tomato Spotted Wilt Orthotospovirus (TSWV)
Review Comments:
The study titled "Evaluation of Wild Peanut Genotypes for Resistance Against Thrips and Thrips-Transmitted Tomato Spotted Wilt Orthotospovirus (TSWV)" provides valuable insights into the potential sources of resistance against TSWV in peanut cultivation. Overall, the research is well written, well-structured, and presents clear findings regarding the evaluation of various genotypes' resistance against TSWV and thrips-mediated transmission. Here are a few specific comments and suggestions to improve the paper:
Line 68: Suggest changing to “whereas evidence suggests that TSWV resistance in peanut resides in multiple chromosomes.”
Lines 95-96: Please review the text formation for clarity and correctness.
Line 172: Please clarify the type of thrips colonies used for thrips FD. Were non-viruliferous or viruliferous thrips employed in the study? This information seems missing from the text.
Lines 246-247: In Figure 3, the bar of ValSten1 is marked as "ab," and the control Georgia Green is marked as "bcd." Please check if there are any significant differences between these two groups and address this in the text.
Overall, this study is a commendable contribution to the understanding of TSWV and Thrips resistance in peanuts and provides useful information for peanut resistance breeding. I suggest accepting it after minor revisions.
Author Response
Dear Reviewer,
My co-authors and I greatly appreciate your comments. We believe that your review and comments have improved our manuscript. We have carefully considered each comment and have tried to address them to the best of our abilities. Our explanation for each comment is included below. Our replies are in bold font. The revisions also are tracked on the manuscript with a red font.
The study titled "Evaluation of Wild Peanut Genotypes for Resistance Against Thrips and Thrips-Transmitted Tomato Spotted Wilt Orthotospovirus (TSWV)" provides valuable insights into the potential sources of resistance against TSWV in peanut cultivation. Overall, the research is well written, well-structured, and presents clear findings regarding the evaluation of various genotypes' resistance against TSWV and thrips-mediated transmission. Here are a few specific comments and suggestions to improve the paper:
Line 68: Suggest changing to “whereas evidence suggests that TSWV resistance in peanut resides in multiple chromosomes.”
Modified as suggested. Currently in lines: 70-71.
Lines 95-96: Please review the text formation for clarity and correctness.
This sentence has now been revised for clarification. Currently in lines: 92-95.
Line 172: Please clarify the type of thrips colonies used for thrips FD. Were non-viruliferous or viruliferous thrips employed in the study? This information seems missing from the text.
They are non-viruliferous thrips, and this has been clarified in line #168.
Lines 246-247: In Figure 3, the bar of ValSten1 is marked as "ab," and the control Georgia Green is marked as "bcd." Please check if there are any significant differences between these two groups and address this in the text.
The mean separation lettering has been corrected, and the text has been appropriately modified in lines 240-241.
Yours sincerely,
Rajagopalbabu Srinivasan

Reviewer 2 Report
The manuscript describes the screening of Arachis germplasm for resistance against TSWV and thrips vectors. Given the limited availability of resistance in cultivate Arachis, wild diploid and induced tetraploid hybrids were screened to evaluate TSWV infection and thrips preference/damage. The work presented is of interest considering the need to identify genetic resources that may help to counteract this very important virus.
A major point that should be considered is indeed the virus itself. In the manuscript the authors always refer to TSWV, without providing information on the isolate(s) in study. Different isolates can however impact the response of a plant genotype; inoculation experiments were also conducted in different years (2019, 2020, 2021) thus the question arise, on the TSWV isolates/populations considered and on consistency for the comparison of the data. In the context of future studies, having information on the virus used for the screenings presented in this specific study would be also valuable, as virus populations may vary overtime. Clarifications on these aspects and further data from the virus characterization would be very valuable to the content of the manuscript.
Comments:
L27-33: In line 26 the order infection, accumulation and injury is used; but thereafter a different order is followed in the abstract. Please harmonize.
L29-30: I don’t see correspondence with the graphs in Fig.5; for example, at 3 weeks, Val seems to have the lowest value; Batdur has the lowest value, but only at 1 week. Please double-check and explain better in relation to time.
L31: if here starts the final summary, I suggest to start the sentence with: Overall, ………….; so that the reader can understand better that this are the final statements
L41: the type species; please mention also the new binomial name
L42: the genome consists of…
L49: “is present”? revise
L68: TSWV resistance in peanut
L80: “The use of Plant Introductions” ? Please clarify
L87: high genetic diversity resistance; can you clarify in the text?
L95-97: check format of References
L113: the authors refer in general to TSWV-infected leaflets, no information is given on the virus; however when referring to resistance, and especially also for comparison of different studies, the characterization of the strain used for screening is a fundamental importance, given the diverse response that strains may have on the host. The experiments were also conducted using infected leaflets collected in different years, but can the TSWV population be considered homogenous to process the data all together? Providing information on the TSWV genome(s) referring to this study would add valuable information, and it is nowadays easily feasible using standard approaches or NGS. The information would be very important also for comparison with future studies.
L125: =>used as control
L147: “transmission efficiency” refers to properties of the transmission; as the authors are referring to percentage of infected plants, the title “Infection rate” could be more appropriate
L148: please add the number of plants used for each line, and the repetitions, as reported in the legend
L178: shouldn’t the formula also take into account the number of leaves per plant? So to normalize the proportion to the number of leaves?
L192: delete “the”
L220: the title refers to wild peanuts, however the data concern also the induced allotetraploids, please revise
Fig.1-2 can be combined: are these local or systemic symptoms (if possible to discriminate); and how many dpi were the pictures taken? Please add in legend
Fig.4: specify in the legend what is A and what B
L301: week should be plural
Fig.6: the behavior of BatSten1 can be correlated to the number of leaves due to defoliation. So the number of scars observed is lower due to this; if this is true, than the formula should be revised to take into account the number of leaves considered. Please clarify in detail.
L315-316: as the resistance to thrips is the important content, please mention here the lines with lowest survival.
L337: it would be good to add the graphs in the main text for immediate understanding
L364-365: re-write the sentence in a less-colloquial way
L382: “wild tetraploids”? please clarify in the context of the other terms used in the paper: “induced allotetraploids” and “cultivated tetraploids”
L381-383: this discussion on the resistance mechanism is not clear and should be argumented better and more extensively.
L384: “identified” is not proper; is associated to
L392: =>showing that a number of…
L400-411: there paragraphs can be better harmonized together in on single section
L412: this was already stated in L405. Please harmonize in the same part of the Discussion.
L427: what is the origin of these 33 species? Please add more information on this collection.
L426: it seems that final remarks start here: pleases start new section
L433: avoid “and… and”
L436: revise “remains”
L445: I think you should refer here to resistance- breaking strains? Not to development of resistance.
L446: “To avoid such a scenario” is too general; is the scenario the use of a limited number of sources of resistance against TSWV?
L444: “at this juncture”????
L449-451: this is already stated in L 442; the concept expressed in lines 441-451 can be better presented to avoid redundancy.
minor edits necessary
Author Response
August 14, 2023
Dear Reviewer,
My co-authors and I greatly appreciate your comments. We believe that your review and comments have improved our manuscript substantially. We have carefully considered each comment and have tried to address them to the best of our abilities. Our explanation for each comment is included below. Our replies are in bold font. The revisions are also tracked on the manuscript with a red font.
The manuscript describes the screening of Arachis germplasm for resistance against TSWV and thrips vectors. Given the limited availability of resistance in cultivate Arachis, wild diploid and induced tetraploid hybrids were screened to evaluate TSWV infection and thrips preference/damage. The work presented is of interest considering the need to identify genetic resources that may help to counteract this very important virus.
A major point that should be considered is indeed the virus itself. In the manuscript the authors always refer to TSWV, without providing information on the isolate(s) in study. Different isolates can however impact the response of a plant genotype; inoculation experiments were also conducted in different years (2019, 2020, 2021) thus the question arise, on the TSWV isolates/populations considered and on consistency for the comparison of the data. In the context of future studies, having information on the virus used for the screenings presented in this specific study would be also valuable, as virus populations may vary overtime. Clarifications on these aspects and further data from the virus characterization would be very valuable to the content of the manuscript.
The point you have raised is valid. It is plausible that different isolates can elicit varying phenotypes. However, that point has been difficult to assess due to a number of limitations that are explained below. A doctoral student is currently sequencing whole genomes of prevalent isolates and phenotyping them on different solanaceous hosts to assess if any isolate-specific host responses are obvious.
Current limitations in the peanut-TSWV pathosystem:
- We have been partially sequencing TSWV isolates from the last ten years including isolates from 2019 used in this study (Sundaraj et al. 2014. Phytopathology. doi: 10.1094/PHYTO-04-13-0107-R; Lai et al. 2021 Pathogens doi: 10.3390/pathogens10111418). The conclusion reached is that there is a tremendous amount of variation in TSWV sequences and therefore a concoction of isolates exists in a given spatial scale. Thus far, purifying selection seems to be the major factor influencing the population genetics and evolution. All isolates seem to induce a very similar phenotype on peanuts.
- Peanut plants are not recalcitrant to TSWV infection (as in producing a hypersensitive response), instead they get systemtically infected with TSWV and display characteristic TSWV-associated symptoms. This seems to be the case in released TSWV-resistant tetraploid cultivars (Shrestha et al. 2013) as well as diploids and allotetraploids in this study. With the characteristic phenotype displayed in every instance, it becomes rather difficult to assess isolate virulence in peanut exclusively. That is precisely why, we are undertaking this effort with other hosts as well.
- Although protocols exist to mechanically inoculate TSWV to peanut, the consistency in replicating this protocol is a problem. Consequently, the ability to maintain the virus mechanically in peanut year-long is not possible. And maintaining the virus in peanut is essential to conduct Frankliniella fusca-mediated TSWV transmission. Also, thrips, Frankliniella fusca, is maintained on peanut to mimic natural settings. This is the main reason why we maintain viruliferous thrips using field-collected foliage.
- Despite these issues, our results have been consistent (Shrestha et al. 2014, Lai et al. 2021). The phenotype induced by TSWV does not seem to be affected by host resistance in tetraploids (Shrestha et al. 2013), and diploids in this study. Also, it should be noted that Georgia Green (TSWV-susceptible control) was used as a control in every transmission assay with diploids and allotetraploids). Variations in virus infection/phenotype severity were always gauged with the susceptible cultivar. No fluctuations in TSWV phenotype severity on Georgia Green was noticed throughout the course of the study.
Comments:
L27-33: In line 26 the order infection, accumulation and injury is used; but thereafter a different order is followed in the abstract. Please harmonize.
The order of infection, accumulation, and feeding injury is followed throughout as suggested.
L29-30: I don’t see correspondence with the graphs in Fig.5; for example, at 3 weeks, Val seems to have the lowest value; Batdur has the lowest value, but only at 1 week. Please double-check and explain better in relation to time.
This information has been corrected as suggested in lines 29-32 in the abstract.
L31: if here starts the final summary, I suggest to start the sentence with: Overall, ………….; so that the reader can understand better that this are the final statements
Modified as suggested. Currently in lines # 32-34.
L41: the type species; please mention also the new binomial name
(Orthotospovirus tomatomacula) included in lines # 43-44.
42: the genome consists of…
Modified as suggested
49: “is present”? revise
Modified as suggested
L68: TSWV resistance in peanut
Modified as suggested
80: “The use of Plant Introductions” ? Please clarify
“The use of Plant Introductions” has been removed and the sentence has now been clarified. Currently in lines 82-84.
L87: high genetic diversity resistance; can you clarify in the text?
Has been modified as ‘high genetic diversity’, currently in line 87.
L95-97: check format of References
Format has been corrected.
L113:
the authors refer in general to TSWV-infected leaflets, no information is given on the virus; however when referring to resistance, and especially also for comparison of different studies, the characterization of the strain used for screening is a fundamental importance, given the diverse response that strains may have on the host. The experiments were also conducted using infected leaflets collected in different years, but can the TSWV population be considered homogenous to process the data all together? Providing information on the TSWV genome(s) referring to this study would add valuable information, and it is nowadays easily feasible using standard approaches or NGS. The information would be very important also for comparison with future studies.
- We recognize the point you have raised. However, we have tried to highlight some of the limitations above in our reply earlier. It is true that multiple isolates from one or two locations were used to conduct transmission assays.
- There was no indication of variance in virulence to refer to them as highly-virulent strains or varying strains. This effort has been clearly documented in our earlier studies (Sundaraj et al. 2014, Lai et al. 2021). In this situation, in exclusively working with peanut, it may be impossible to associate phenotype-isolate characteristics.
- Moreover, what is interesting is that the results across the years were congruent and consistent, reiterating that isolate-host phenotypes were rather uniform.
- Also, it should be noted that Georgia Green (TSWV-susceptible control) was used as a control in every transmission assay with diploids and allotetraploids). Variations in virus infection/phenotype severity were always gauged with the susceptible cultivar. No fluctuations in TSWV phenotype severity on Georgia Green was noticed throughout the course of the study.
- We do agree with you in that assessing host phenotype-strain/isolate differences is very important; however, that might not be possible exclusively with peanut (especially due to lack systemic infection as opposed to hypersensitive resistance in Solanaceae hosts).
- That is why we are undertaking the effort led by doctoral student with multiple hosts. Also, early results from NGS-based TSWV genome sequencing have been less than ideal. This is mainly due to issues with sequencing non-coding and intergenic regions. We have been resorting more towards a Sanger sequencing approach consequently. Nevertheless, your point is valid, we have acknowledged it in the discussion section, and we are working towards addressing the same in the near future.
L125: =>used as control
Modified as suggested. Currently in lines # 122-123.
147: “transmission efficiency” refers to properties of the transmission; as the authors are referring to percentage of infected plants, the title “Infection rate” could be more appropriate
Has now been modified as ‘incidence of TSWV infection’ in line 142.
L148: please add the number of plants used for each line, and the repetitions, as reported in the legend
This information is included in lines 137-138 as well as 195-197.
L178: shouldn’t the formula also take into account the number of leaves per plant? So to normalize the proportion to the number of leaves?
Yes, you are correct. The proportion here refers to the number of leaflets with feeding scars/total number of leaflets. This has been modified in lines 173-174.
L192: delete “the”
Deleted as suggested
220: the title refers to wild peanuts, however the data concern also the induced allotetraploids, please revise
The title has now been modified “Evaluation of Wild Peanut Species and their Allotetraploids for Resistance Against Thrips and Thrips-Transmitted Tomato Spotted Wilt Orthotospovirus (TSWV)”
Fig.1-2 can be combined: are these local or systemic symptoms (if possible to discriminate); and how many dpi were the pictures taken? Please add in legend
Figures have been combined as suggested. The pictures were taken approximately three weeks post inoculation.
Fig.4: specify in the legend what is A and what B
Clarified both on the figure as well as the legend.
L301: week should be plural
Corrected as indicated.
Fig.6: the behavior of BatSten1 can be correlated to the number of leaves due to defoliation. So the number of scars observed is lower due to this; if this is true, than the formula should be revised to take into account the number of leaves considered. Please clarify in detail.
In every instance the feeding damage index (now Fig 5) included the proportion of leaflets with feeding scars. In each of these instances, the proportion was calculated as number of leaflets with feeding scars over the total number of leaflets. The drop observed with this genotype could be due to a greater number of leaflets/leaves with scars dropping versus the total number of non-scarred leaves dropping, and the reduced feeding severity on the remaining leaves contributed to the drop in the index noticed.
L315-316: as the resistance to thrips is the important content, please mention here the lines with lowest survival.
The survival of thrips feeding on diploids was five times lower than in the control, Georgia Green. Similar to the diploids, the number of adults emerged on tetraploids was lower than in the control. Within the tetraploid genotypes, thrips survival was the highest in MagSten1 and lowest in BatDur1. Currently in lines 311-314.
L337: it would be good to add the graphs in the main text for immediate understanding
Included as suggested.
L364-365: re-write the sentence in a less-colloquial way
Has been appropriately modified in lines 370-373 “whereas this study used thrips-mediated inoculation. Thrips-mediated inoculation has shown to be more effective and consistent than mechanical inoculation. Lai et al. (2015), also used thrips-mediated inoculation and demonstrated that A. diogoi V10602 accession can get infected with TSWV.”
L382: “wild tetraploids”? please clarify in the context of the other terms used in the paper: “induced allotetraploids” and “cultivated tetraploids”
Usage of ‘wild tetraploids’ has been appropriately edited.
L381-383: this discussion on the resistance mechanism is not clear and should be argumented better and more extensively.
This sentence has been modified and synchronized with the others below to elaborate plausible TSWV resistance mechanism. Currently in lines 398-406.
L384: “identified” is not proper; is associated to
Modified as suggested. Line 406.
L392: =>showing that a number of…
Has been modified.
L400-411: there paragraphs can be better harmonized together in on single section
The three paragraphs have now been synthesized better and are included in a single paragraph as suggested. Currently in lines 418-437.
L412: this was already stated in L405. Please harmonize in the same part of the Discussion.
Modified as suggested.
L427: what is the origin of these 33 species? Please add more information on this collection.
The number 33 might not be an accurate count of the number of available Arachis wild species. We have now resorted to not using that number and instead use ‘many’. Line 444.
L426: it seems that final remarks start here: pleases start new section
Modified as suggested.
L433: avoid “and… and”
Modified as suggested
L436: revise “remains”
Modified as suggestedL445: I think you should refer here to resistance- breaking strains? Not to development of resistance.L446: “To avoid such a scenario” is too general; is the scenario the use of a limited number of sources of resistance against TSWV?L444: “at this juncture”????L449-451: this is already stated in L 442; the concept expressed in lines 441-451 can be better presented to avoid redundancy.
Entire section has been revamped to incorporate your suggestions and avoid redundancy.
Thanks once again for your thorough review, we appreciate it. We look forward to hearing from you.
Yours sincerely,
Rajagopalbabu Srinivasan

Round 2
Reviewer 2 Report
I am grateful for the authors for the effective response to the comments and the effort to revise sections of the paper where clarification was suggested. The manuscript has been improved consistently. The main point regarding the TSWV isolates has been now explained and put in the context of the TSWV/peanut pathosystem. Indeed the fact that that the results across the years were congruent and consistent justifys that even in presence of high TSWV diversity the response is comparable. However, I still remain a big fan of providing information on the TSWVs used in the study, and if the authors have even partial sequences in addition to the ones in the 2019 manuscript, or if they could sequence the material from later years 2020-2021 used in the experiments and eventually stored, this would be very valuable. This effort would be not in the context of different symptoms by different isolates, rather to provide information on the TSWV strains/diversity in the study. I understand this is quite an affort and the authors are already conducting a new study on this, however I would like to remark this to the authors again as virus isolate(s) information would be significant, and as well I acknowledge the additional information added on this specific point in the Discussion.
Author Response
August 23, 2023
Dear Reviewer,
We appreciate your comments and suggestion. My co-authors have partially sequenced TSWV peanut isolates from 2020 and 2021, and the details of the sequences are in the supplementary file. The sequences have been submitted to NCBI GenBank as well. Once we receive the accession numbers, they will be incorporated into the manuscript.
Thanks once again for your thorough review, we appreciate it. We look forward to hearing from you.
Yours sincerely,
Rajagopalbabu Srinivasan
(Professor)
